# Vaccination with VLPs Presenting a Linear Neutralizing Domain of *S. aureus* Hla Elicits Protective Immunity

**DOI:** 10.3390/toxins12070450

**Published:** 2020-07-11

**Authors:** Jason A. Joyner, Seth M. Daly, Julianne Peabody, Kathleen D. Triplett, Srijana Pokhrel, Bradley O. Elmore, Diane Adebanjo, David S. Peabody, Bryce Chackerian, Pamela R. Hall

**Affiliations:** 1Department of Pharmaceutical Sciences, College of Pharmacy, University of New Mexico Health Sciences Center, Albuquerque, NM 87131, USA; JAJoyner@salud.unm.edu (J.A.J.); SDaly@salud.unm.edu (S.M.D.); KTriplett@salud.unm.edu (K.D.T.); SPokhrel@salud.unm.edu (S.P.); bradleyelmore@gmail.com (B.O.E.); 2Department of Molecular Genetics and Microbiology, School of Medicine, University of New Mexico Health Sciences Center, Albuquerque, NM 87131, USA; JPeabody@salud.unm.edu (J.P.); DtAdebanjo@salud.unm.edu (D.A.); DPeabody@salud.unm.edu (D.S.P.); BChackerian@salud.unm.edu (B.C.)

**Keywords:** *Staphylococcus aureus*, α-hemolysin, linear neutralizing domain, virus-like particles, vaccine, mice

## Abstract

The pore-forming cytotoxin α-hemolysin, or Hla, is a critical *Staphylococcus aureus* virulence factor that promotes infection by causing tissue damage, excessive inflammation, and lysis of both innate and adaptive immune cells, among other cellular targets. In this study, we asked whether a virus-like particle (VLP)-based vaccine targeting Hla could attenuate *S. aureus* Hla-mediated pathogenesis. VLPs are versatile vaccine platforms that can be used to display target antigens in a multivalent array, typically resulting in the induction of high titer, long-lasting antibody responses. In the present study, we describe the first VLP-based vaccines that target Hla. Vaccination with either of two VLPs displaying a 21 amino-acid linear neutralizing domain (LND) of Hla protected both male and female mice from subcutaneous Hla challenge, evident by reduction in lesion size and neutrophil influx to the site of intoxication. Antibodies elicited by VLP-LND vaccination bound both the LND peptide and the native toxin, effectively neutralizing Hla and preventing toxin-mediated lysis of target cells. We anticipate these novel and promising vaccines being part of a multi-component *S. aureus* vaccine to reduce severity of *S. aureus* infection.

## 1. Introduction

*Staphylococcus aureus* α-hemolysin (Hla) is an important secreted bacterial virulence factor whose loci is found in 99% of clinical isolates. Hla mediates invasive infection and promotes pathogenesis associated with both primary and recurrent skin and soft tissue infection (SSTI), pneumonia (PNA), peritoneal infections, and sepsis, among others [1,2,3,4,5,6,7,8,9]. In SSTI models, *S. aureus* mutants lacking Hla are attenuated [10] and are more rapidly cleared by the host [3]. Hla binds to a zinc metalloprotease, ADAM10, on host cells to form a heptameric pore and initiate breach of epithelial barriers [6,9,11]. The importance of Hla to various *S. aureus* infections likely stems from the broad cellular distribution of ADAM10 [7]. Therefore, Hla is a major toxin target for vaccines and therapeutics to limit *S. aureus* infections.

Several Hla vaccines have been tested in preclinical animal models including (i) a full length nontoxigenic Hla (Hla_H35L_), (ii) the N-terminal 50 amino acids of Hla fused to glutathione S-transferase (GST) (GST-Hla_1-50_), (iii) a structurally designed vaccine consisting of 62 non-contiguous Hla amino acids, and (iv) Hla engineered to lack the predicted membrane-spanning stem domain (HlaPSGS) [10,12,13,14]. Despite some successes in animal models, no *S. aureus* or Hla vaccine has been successful in clinical trials. This, together with the burden of disease caused by *S. aureus*, suggests that novel approaches to vaccine development are urgently needed.

Virus-like particles (VLPs) are a versatile nanoparticle vaccine platform for displaying practically any epitope in a multivalent format that virtually guarantees the ability to elicit high titer and long-lasting epitope-specific Abs [15]. While many VLP platforms can be used to display target antigens, VLPs derived from spherical RNA bacteriophage, such as Qß and AP205, have many advantages, including the ability to express and purify large amounts of VLPs in *E. coli*, and the multiple flexible approaches for displaying antigenic targets on their surface. Several Qß VLP-based vaccines have been evaluated in clinical trials, and these vaccines were shown to be safe and immunogenic [16]. Although VLP-based vaccines targeting *S. aureus* toxins have yet to be developed, their successful utilization against other pathogens suggests their potential for vaccine protection of humans against Hla-mediated pathogenesis. We developed active VLP-based vaccines by displaying a 21 amino-acid Hla linear neutralizing domain (LND), first identified by Oscherwitz and Cease as the target of an Hla-inactivating mAb [17]. The LND domain is involved in heptamerization of the Hla (Figure 1A), and it has been shown that an antibody against this epitope can neutralize Hla activity. We postulated that vaccination with VLPs displaying this peptide would elicit a neutralizing antibody (NAb) response and provide active protection in a mouse model of *S. aureus* Hla challenge.

To test our postulate, we vaccinated mice with two different VLPs displaying the Hla-LND and assessed vaccine efficacy using a murine skin challenge model. Here, we demonstrate that vaccination with LND-VLPs induces Hla-reactive antibodies that provide protection against lesion formation upon subcutaneous challenge with recombinant Hla in both male and female mice. In addition, these Abs prevented Hla-mediated lysis of Jurkat cells in an in vitro neutralization assay. Together, our findings demonstrate the efficacy of VLP-based vaccines displaying the Hla-LND and suggest that these vaccines could contribute to a multi-component vaccine to prevent *S. aureus* pathogenesis and infection.

## 2. Results

### 2.1. Vaccination with VLPs Displaying LND Protect against Hla Challenge 

We used two different techniques for displaying the 21 amino acid Hla-LND epitope (Figure 1A) [17] on VLPs. First, we produced recombinant VLPs by genetically fusing this epitope to the C-terminus of the bacteriophage AP205 coat protein (AP205-LND; Figure 1B). This technique has been used previously to generate recombinant AP205 VLPs [18]. AP205 VLPs consist of 180 copies of the viral coat protein; thus, each VLP displays 180 copies of Hla-LND. As an alternative strategy, we chemically conjugated a synthetic peptide representing LND to Qß VLPs using a bifunctional crosslinker (Figure 1C). Formation of intact VLPs, purity of the preparations, and conjugation efficiency was determined by agarose gel electrophoresis and SDS-PAGE (Appendix A). Based on the efficiency of LND conjugation, we determined that Qβ VLPs display an average of approximately 190 copies of the LND peptide per particle. 

To assess the efficacy of our VLP-LND vaccines, we used a well-established mouse model of *S. aureus* skin infection/intoxication, where formation of a necrotic lesion is Hla-dependent [10,19,20,21]. In this model, the necrotic lesion visibly forms on the skin surface and lesion size is easily measurable [22,23,24,25,26], allowing straightforward assessment of Hla vaccine efficacy. Mice were immunized with two doses of AP205-LND or Qβ-LND VLPs. As controls, mice were immunized with similar doses of wild type, unmodified VLPs or were mock vaccinated with PBS. Eight weeks post-boost, mice were challenged by subcutaneous injection of rHla. The ability of vaccination to protect mice was evaluated by measuring lesion area for three days after Hla injection and by the calculated area under the curve (AUC) for the three-day challenge. Male and female mice were analyzed separately because female mice are more resistant to Hla compared to males [23], and different concentrations of Hla were subcutaneously injected in effort to achieve similar size lesions between the sexes. Hla-induced lesion size was significantly reduced in both AP205-LND and Qβ-LND vaccinated male (~60% and 94%, respectively) and female mice (~79% and 73%, respectively) compared to control mice (Figure 2A–C). The two VLP-based vaccines provided statistically similar protection from lesion formation in both sexes. 

Along with reduced lesion size, we predicted that in vivo Hla challenge of VLP-LND-vaccinated mice would result in reduced local neutrophil influx and reduced inflammation. To assess inflammatory responses, we conducted histological analysis of lesions collected from vaccinated mice on day three post-Hla challenge. While control-vaccinated mice displayed a neutrophilic barrier crossing the dermis and epidermis surrounding the intoxication site, lesions from VLP-LND-vaccinated mice exhibited reduced numbers of neutrophils influx and decreased tissue damage (Figure 2D). This indicates that VLP-LND vaccination limits the pro-inflammatory response to Hla challenge.

### 2.2. Antibodies Induced by VLP-LND Vaccination Bind to Intact Hla

In order to assess the mechanism(s) of in vivo protection, we asked whether serum Abs elicited by VLP-LND vaccination could bind Hla protein in vitro. As shown in Figure 3, vaccination with AP205-LND and Qβ-LND VLPs elicited high titer anti- rHla IgG antibodies, whereas no rHla-binding was detected in control mice (Figure 3A). These data indicate that both VLP-LND vaccines induce Ab recognizing Hla, suggesting that the conformation of the LND epitope displayed on the surface of the VLP is sufficiently similar to that of the target domain of the full-length toxin. 

To investigate the molecular mimicry of the LND peptide in the context of AP205 and Qβ VLPs to the target domain of the full-length toxin, we also compared the ability of mouse VLP-LND antisera to bind to a synthetic LND peptide versus full-length rHla. The ratio of IgG Ab titers to the LND peptide versus rHla was not statistically different between mice vaccinated with AP205-LND or Qβ-LND, with ratios averaging 1.11 versus 1.13 respectively (Figure 3B). These findings support our hypothesis that the LND epitope displayed on either VLP platform exhibits molecular mimicry to the target epitope in Hla. 

### 2.3. AP205-LND Vaccination Induces Hla-Neutralizing Antibodies

To determine the Hla-neutralizing ability of Ab elicited by VLP-LND vaccination, we performed a toxin neutralization assay as previously described [17]. Sera collected from vaccinated mice was tested for its ability to prevent lysis of susceptible Jurkat cells, a human T-cell line. AP205-LND antisera, but not sera raised against wild type AP205 VLPs, potently inhibited Jurkat cell lysis (Figure 4A) indicating that vaccination elicited Abs that neutralize Hla activity. 

We next asked whether lesion size in individual Hla-challenged mice inversely correlated with Hla antibody titer. To test this, we performed a Hla ELISA on individual sera collected from vaccinated, Hla challenged mice at the time of sacrifice. Lesion area inversely correlated with the log of α-Hla titer for individual mice vaccinated with VLP-LND (*p* < 0.007; Figure 4B). For example, the individual mouse with the lowest anti-Hla antibody titer also had the largest legion after Hla-challenge. Finally, we tested whether antibodies were present at the site of Hla-challenge by detecting the presence of anti-Hla IgG in lesion homogenates by ELISA (Figure 4C). Anti-Hla IgG antibodies were readily detectable at the lesion site of VLP-LND immunized mice, but not controls, indicating that antibodies are present at the site of challenge. Together, these findings point to an in vivo mechanism of action whereby VLP-LND vaccination elicits an antibody response to *S. aureus* Hla, infiltration of anti-Hla Ab to the site of intoxication, and neutralization of Hla function, thus providing protection against Hla-mediated pathogenesis. 

## 3. Discussion

Oscherwitz and Cease [17] previously demonstrated the ability of a linear 21 amino-acid region of Hla, termed the linear neutralizing domain (LND), to elicit Hla NAbs. We utilized this region to develop VLP-based vaccines that protect against Hla challenge in a murine model of SSTI. VLP-based vaccines, including some already in clinical use, have historically been developed against viral pathogens [16], making their utilization to protect against bacterial toxins a novel strategy. We used two strategies to display the LND epitope on VLPs; genetic fusion to AP205 and chemically conjugation to Qß. Both VLPs displayed the LND peptide at a similar valency (180–190 copies per VLP) and in a relatively unconstrained format on the surface of the VLP. Correspondingly, both VLP-based vaccines elicited high-titer antibodies to both the LND and to intact Hla and conferred protection against Hla skin challenge in a murine model. Furthermore, AP205-LND-elicited Abs neutralized Hla function and prevented lysis of cultured Jurkat cells. Finally, Hla-targeting Abs were found at the site of Hla challenge in the mouse model and the Hla titer inversely correlated with lesion size. Together, these findings point to an in vivo mechanism of action whereby VLP-LND vaccination elicits an antibody response capable of neutralizing *S. aureus* Hla, thus providing protection against Hla-mediated pathogenesis. 

Although opsonizing antibodies presumably have the potential to clear *S. aureus* infection [27,28,29], rational design of *S. aureus* vaccines aimed at generating opsonizing antibodies have so far been clinically unsuccessful [30]. These failures suggest that an alternative approach that targets virulence factors like Hla could be effective at reducing pathogenesis or could be effective in the context of a combination vaccine. Hla and other *S. aureus* pore-forming toxins are capable of lysing host immune cells [7] and contribute to several types of *S. aureus* infection [1,2,3,9]. Neutralization of pore-forming toxins could therefore help preserve phagocytic cells such as neutrophils and monocytes/macrophages, thus secondarily enhancing bacterial clearance [30]. The importance of preserving phagocyte numbers is evidenced by the higher incidence of *S. aureus* infection in patients with functional defects in these cells [31,32,33]. In addition, human clinical data indicate that a high Ab titer against Hla correlates with decreased severity of *S. aureus* sepsis [34] and reduced recurrence of *S. aureus* SSTI in children [35]. 

Hla promotes invasive infection and is therefore a target for therapeutic and prophylactic interventions. Although Hla-targeting vaccines have shown promise in animal models of pneumonia [12,13,36], SSTI [10,17,37,38], and bacteremia [13], even vaccines inducing opsonophagocytic Abs show efficacy in animal models, but subsequently failed in human clinical trials [39]. The VLP-based vaccines reported here focus on a targeted, essential domain of Hla and thus direct the immune response towards a vulnerable epitope while avoiding non-neutralizing regions present in the native toxin. It is worth noting that many of the other pore forming toxins secreted by *S. aureus* also encode what appears to be linear neutralizing domains. Similar to what we have shown here regarding the Hla-LND, it may be possible to target the LNDs of other *S. aureus* pore-forming toxins with VLP-based vaccines that could have efficacy against a variety of *S. aureus* infections. Therefore, vaccines that elicit NAbs against pore-forming toxins, such as Hla, have the potential to limit pathogenesis and enhance bacterial clearance by sparing host immune cells.

## 4. Conclusions

Novel treatment and prevention strategies are urgently needed to combat infections caused by antibiotic resistant *S. aureus*. Here, we describe two candidate VLP-based vaccines, AP205-LND and Qβ-LND, with demonstrated efficacy in the prevention of Hla-mediated pathogenesis in a murine skin intoxication model. Although further studies are required, the efficacy of these vaccines and the clinical utility of VLP-based vaccines in general suggests that they could contribute to a multi-component *S. aureus* vaccine. 

## 5. Materials and Methods 

### 5.1. Ethics Statement

Animal experiments described in this study were approved (3/25/2020) by the Institutional Animal Care and Use Committee (IACUC) of the University of New Mexico Health Sciences Center (Animal Welfare Assurance number D16-00228) and conducted in strict accordance with recommendations in the *Guide for Care and Use of Laboratory Animals* [40] the Animal Welfare Act and U.S. federal law. 

### 5.2. VLP Cloning, Expression, and Purification

The pAPKP plasmid encoding the AP205 coat protein and C-terminal linker (GTAGGGSGT) was used for generation of AP205-LND. The pAPKP plasmid was derived from pBAD-thio-TOPO-AP205, provided by Kaspars Tars of the Latvian Biomedical Research and Study Centre, Riga. It is based on pBAD-thio-TOPO (ThermoFisher, Grand Island, NY, USA) and expresses AP205 coat protein from the *E. coli* arabinose promoter. We constructed pAPKP by modifying pBAD-thio-TOPO-AP205 to encode a linker peptide at the coat protein C-terminus similar to that described by Tissot et al. in 2010 [18]. Briefly, the *E. coli* optimized Hla-LND sequence was synthesized as a mini-gene in IDT’s pSMART plasmid (Integrated DNA Technologies, Coralville, IA, USA). We fused foreign peptides to coat protein through the linker by insertion of their coding sequences between the KpnI and PstI sites of pAPKP and the LND-insert. Complete plasmid sequences are available upon request. The pAPKP-Hla-LND was transformed into *E. coli* 5αF’I^q^ for transcription followed by transformation into *E. coli* BL21*DE3 for expression.

Wild type Qβ was expressed from plasmid pETQCT and purified from *Escherichia coli* (*E. coli*) as previously described [41,42]. The synthetic Hla-LND peptide (Hla 119-131; GFNGNVTGDDTGKIGGLIGAN), including an N-terminal cysteine residue followed by a three-glycine-spacer sequence (CGGG-LND), was synthesized by GenScript (Piscataway, NJ, USA) then conjugated to Qβ’s surface-exposed lysine residues using the bifunctional crosslinker succinimidyl 6-[(beta-maleimidopropionamido) hexanoate] (SMPH; 22363; Thermo Scientific, Waltham, MA, USA) [43].

Wild type AP205 and AP205-LND were expressed and purified from BL21*DE3. Overnight cultures with 50 ug/mL kanamycin (Sigma-Aldrich, Burlington, MA, USA) were grown in 2xYT (16 g/L tryptone, 10 g/L yeast extract, 5 g/L NaCl; BD Biosciences, Franklin Lakes, NJ, USA). Expression strains of both AP205 and AP205-LND were grown overnight on media with kanamycin and used to inoculate a 50 to 500 mL culture with shaking. When OD_600_ nm surpassed 0.6, the culture was induced with 0.2% *w/v* arabinose and incubated for an additional 3 h. The cells were pelleted by centrifugation and purified via FPLC and confirmed for purity as previously described [25].

### 5.3. Mouse Immunizations

Four-week-old male and female BALB/c mice (Jackson Laboratories, Bar Harbor, ME, USA) were immunized by injection of 50–60 μL vaccine mixture in the caudal thigh muscle. Vaccines contained 10 μg VLP suspended in sterile PBS, mixed 1:1 in Incomplete Freund’s Adjuvant (Invivogen, San Diego, CA, USA), or mock vaccinated with PBS alone. Four weeks later, mice were given an identical vaccination (boost). Serum for ELISAs was collected by retro-orbital bleed without euthanizing or at the time of euthanasia by cardiac puncture. Challenge experiments were performed eight weeks after boosting. 

### 5.4. Hla and LND Peptide ELISA

Vaccine-elicited murine serum antibody binding to Hla was determined by ELISA. ELISA plates were prepared by coating Ultra Cruz ELISA High-Binding plates (Santa Cruz Biotechnology, Santa Cruz, CA, USA) with 125 ng per well of rHla in 50 μL carbonate-bicarbonate buffer (pH 9.5) and incubated overnight (approximately 16–18 h) with mild shaking at 4 °C. After incubation, liquid was removed, and wells were blocked with PBS containing 1% casein (Thermo Scientific, Waltham, MA, USA) + 0.05% Tween-20 and then incubated for 2 h at 22 °C with shaking. Sera from vaccinated mice were serially diluted 2-fold or 4-fold for control samples, added onto the Hla-coated wells for antibody binding to Hla, and incubated at 22 °C for 1 h with shaking. After serum Ab binding, excess liquid was removed, and plates were washed three times using PBS containing 0.05% Tween-20 (PBS-T). Murine Ab bound to Hla was detected using polyclonal goat anti-mouse IgG secondary antibody linked to HRP (32230, Invitrogen), and assays were developed using 1-Step^TM^ Ultra TMP-ELISA according to manufacturer’s directions. Hla titer was calculated as the reciprocal of the dilution value of the furthest diluted well with absorbance greater than two times the average of background (naïve sera). The lower limit of detection (LLOD) was a titer of 100. For detection of anti-Hla IgG Ab in Hla lesions, lesion homogenates collected from Hla-challenged mice, as reported below, were used in place of serum in the Hla ELISA.

For the LND peptide ELISA, streptavidin was coated to Immulon II ELISA plates (Thermo Scientific, Waltham, MA, USA) at 500 ng per well in pH 7.4 PBS overnight at 4 °C. Unbound streptavidin was removed by washing with PBS, and then, SMPH (22363; Thermo Scientific, Waltham, MA, USA) was added at 1 µg per well and incubated for 1 h at room temperature with shaking. After washing with PBS, 1 µg of synthetic CGGG-LND peptide (Genscript, Piscataway, NJ, USA) was added to each well and incubated for 1 h at room temperature with shaking. Unconjugated peptide was removed by washing with PBS, plates were blocked with PBS + 0.5% non-fat dry milk, and the ELISA was conducted as described above.

### 5.5. Toxin Neutralization Assay

Vaccine-elicited Hla-neutralizing Ab were detected in vitro by toxin neutralization assay, similar to that described by Oscherwitz and Cease [17] using the Jurkat human T cell line (TIB-152, ATCC, Manassas, VA, USA). Jurkat cells were grown in RPMI 1640 (Life Technologies, Grand Island, NY, USA) supplemented with 10% fetal bovine serum and 50 μM BME (complete medium) in a humidified 5% CO_2_ incubator at 37 °C. Fifty μL of vaccinated mouse sera diluted 1:12 in complete medium was plated in triplicate in a 96-well polystyrene plate (Corning Incorporated, Corning, NY, USA) and incubated with 50 μL rHla at 0.8 μg/μL for 30 min at 37 °C. Approximately 100,000 Jurkat cells in 100 μL complete medium were combined with the sera-rHla mixture and incubated at 37 °C. After 2 h incubation, 100 μL of 2,3-bis-(2-methoxy-4-nitro-5-sulfophenyl)-2H-tetrazolium-5-carboxanilide (XTT; Sigma-Aldrich, St. Louis, MO, USA) at 0.3 mg/mL and phenazine methosulfate (PMS; TCI America, Portland, OR, USA) at 0.015 mg/mL diluted in complete medium was added to the wells. After 18 h incubation, absorbance at 450 nm was measured using a SpectraMax 340 microplate reader (Molecular Devices, San Jose, CA, USA). As a positive control, 8B7 Hla-neutralizing antibody (0210-001, IBT, Rockville, MD, USA) was added to naïve mouse sera. 

### 5.6. Mouse Skin Intoxication Model

The murine model for Hla-mediated lesion formation was essentially performed as previously described [19]. Briefly, the right flank of vaccinated male or female BALB/c mice was cleared of hair by shaving and depilation two days prior to intoxication. Because female mice are more resistant to Hla compared to males [23], in an effort to achieve similar size lesions between the sexes, male mice were challenged with 1.25 µg of Hla while females were challenged with 2 µg. On the day of intoxication, rHla was diluted in 50 μL saline (Braun, Irvine, CA, USA). Mice were anesthetized by isoflurane inhalation and injected subcutaneously with rHla in saline. Mice were weighed on the day of injection and daily thereafter until sacrifice. The site of intoxication was photographed daily, and lesion size was determined by analysis with ImageJ^11^. Three days after intoxication, mice were sacrificed by CO_2_ asphyxiation, and a 225-mm^2^ skin section containing the lesion was excised and stored in Hank’s Balanced Salt Solution (HBSS; Life Technologies, Carlsbad, CA, USA) containing 10 mM HEPES. Lesions were homogenized by mechanical disruption, and the clarified fraction was stored at −80 °C with 1x Halt^TM^ protease inhibitor (Thermo Scientific, Waltham, MA, USA) for further analysis.

### 5.7. Histology

Histological evaluation was performed as previously reported [23], with skin samples collected on day 3 post-Hla intoxication. A 225-mm^2^ piece of skin containing the site of Hla injection was flattened on cardboard, fixed for 24 h in neutral buffered formalin, and then stored in 70% ethanol. Fixed skin was bisected through the center of the injection site and processed for embedding in paraffin, sectioned at 5 μm thickness, and stained with hematoxylin and eosin (H&E). Slides of H&E stained skin sections were scanned by an Aperio CS2 scanner (Leica Biosystems, Buffalo Grove, IL, USA).

### 5.8. Statistical analyses

Statistical analyses were performed using GraphPad Prism 8.4.2. Differences were considered statistically significant at *p* < 0.05.

## Figures and Tables

**Figure 1 toxins-12-00450-f001:**
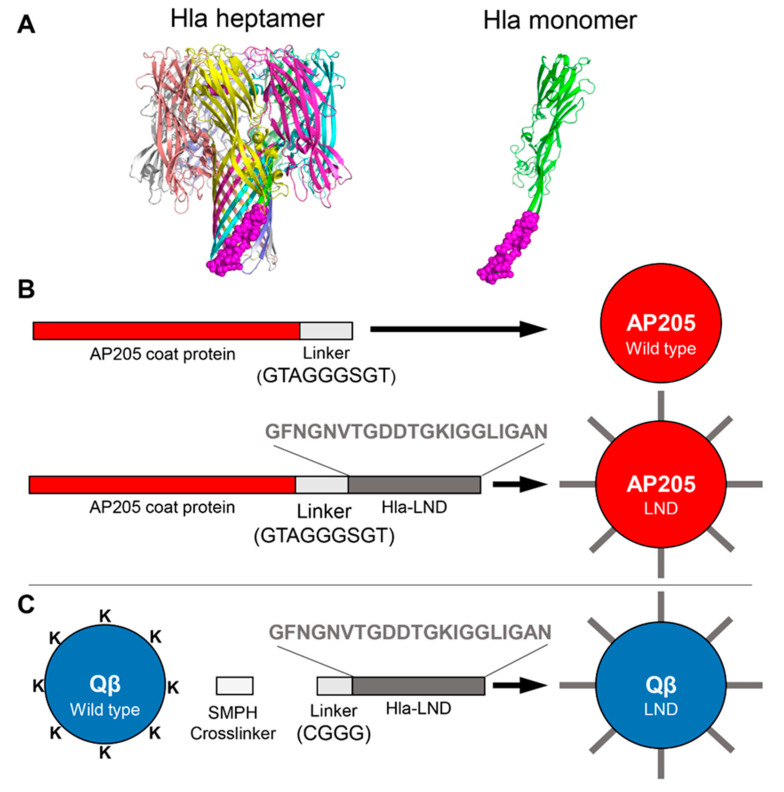
Schematic of virus-like particles (VLPs) Displaying α-hemolysin (Hla) linear neutralizing domain (LND). (**A**) (Left) Ribbon depiction of Hla heptameric pore based on 3ANZ.pdb. Monomers are shown in different colors and LND region shown as purple spheres. (Right) Ribbon depiction of Hla monomer with LND region shown as described above. Figures produced using PyMOL (The PyMOL Molecular Graphics System, Version 2.0 Schrödinger, LLC.) (**B**) (Top left) Linear schematic depicting wild type AP205 coat protein with C-terminal linker; (top right) schematic of assembled AP205 wild type VLP; (Bottom left) linear schematic of AP205 coat protein with Hla-LND sequence genetically inserted; (bottom right) schematic of assembled AP205-LND VLP created through molecular cloning. (**C**) (Left) schematic of assembled Qβ wild type VLP depicting surface exposed lysines; (center) linear depiction of SMPH crosslinker and synthetic CGGG-Hla-LND prior to chemical conjugation to surface lysines; (right) schematic of assembled Qβ VLP displaying surface lysine conjugated LND peptides.

**Figure 2 toxins-12-00450-f002:**
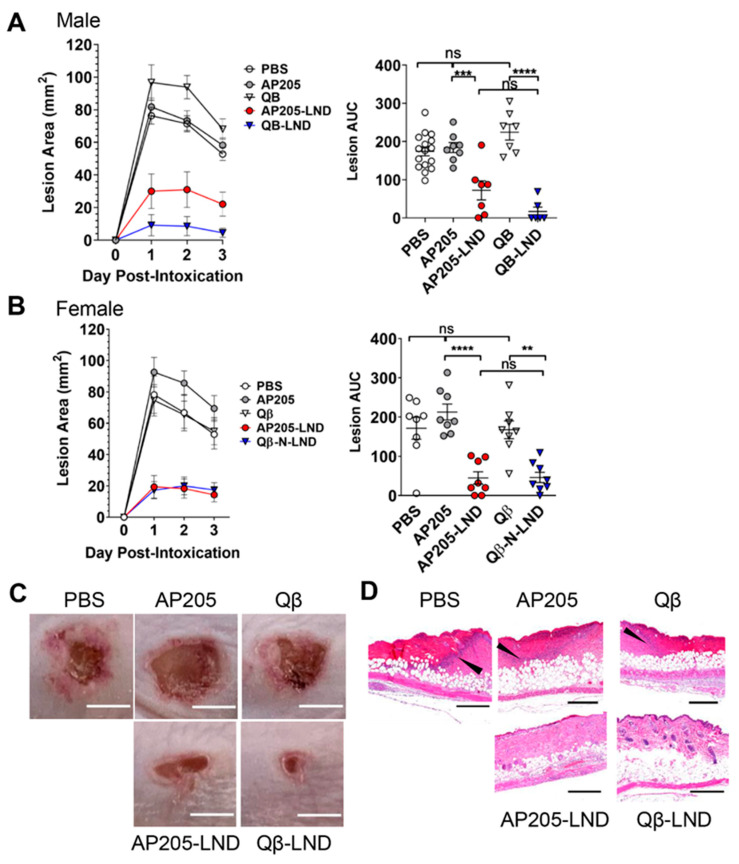
Vaccination with VLPs displaying LND protects mice from lesion formation caused by subcutaneous Hla challenge. (**A**) (Left) Lesion area over the course of a 3-day Hla intoxication of vaccinated or mock-vaccinated male BALB/c mice (1.25 µg Hla/mouse). (Right) Calculated lesion area under the curve (AUC) for the 3-day Hla challenge. (**B**) (Left) Lesion area over the course of a 3-day Hla intoxication of female BALB/c mice (2 µg Hla/mouse). (Right) Calculated lesion AUC for the 3-day Hla challenge. (**C**) Representative images of external lesions of vaccinated Hla-challenged male BALB/c mice at day 3 post-intoxication. Scale bar = 5 mm. (**D**) H&E stained Hla-intoxicated skin sections collected 3-days post-intoxication from vaccinated male BALB/c mice. Arrowheads denote neutrophilic barrier around intoxication site. Scale bar = 0.5 mm. (**A**,**B**) Data are mean ± SEM for (**A**) n = 16, 8, 7, 7, 6 mice per group and (**B**) n = 8 mice/group. One-Way ANOVA *p* < 0.0001, Tukey’s multiple comparisons test; ns, not significant; **, *p* < 0.01, ***, *p* < 0.001, ****, *p* < 0.0001.

**Figure 3 toxins-12-00450-f003:**
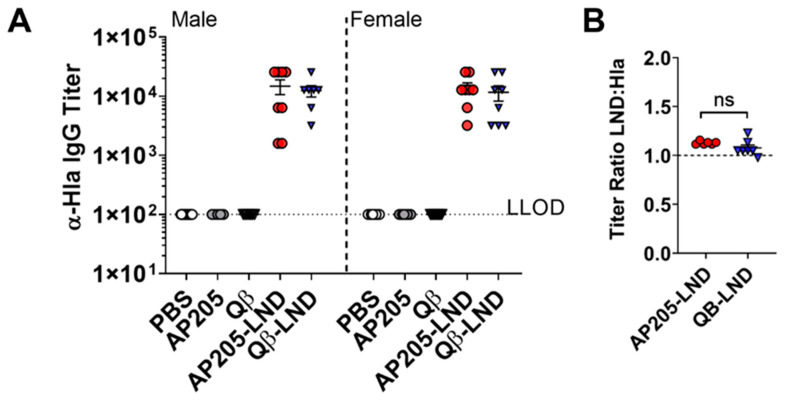
Antibodies elicited by VLP-LND vaccination recognize recombinant Hla. (**A**) Antibody binding to Hla from serum from male (n = 5 to 16 mice/group) or female (n = 8 mice/group) BALB/c mice vaccinated with AP205-LND, Qβ-LND or controls, at two weeks post-boost, tested by ELISA. LLOD = lower limit of detection. (**B**) Ratio of anti-LND antibody titer over anti-Hla titer using serum from male BALB/c mice vaccinated with AP205-LND (n = 8 mice/group) or Qβ-LND (n = 7 mice/group). All antibody titers are reported as the reciprocal of the highest serum dilution with an OD450 2-fold greater than naïve sera. Data shown as mean ± SEM. ns, not significant.

**Figure 4 toxins-12-00450-f004:**
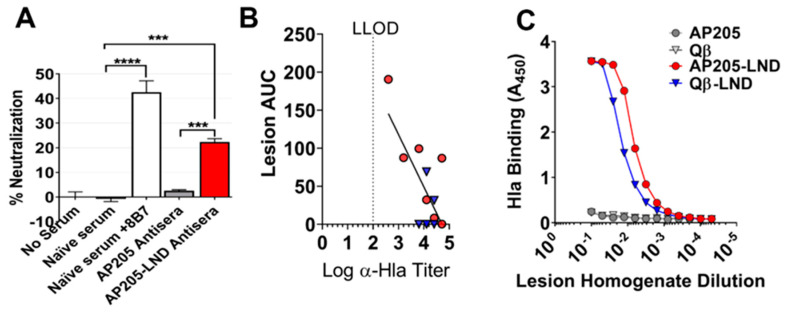
VLP-LND vaccination elicits antibodies that protect against Hla-mediated pathogenesis in vitro and in vivo. (**A**) Toxin neutralization assay measuring viability of Jurkat cells incubated with 0.8 µg/mL Hla combined with sera collected from vaccinated mice diluted 1:12. Sera include pooled naïve BALB/c serum (n = 5 mice/group), naïve serum spiked with ~9 μg/mL Hla-neutralizing mAb 8B7 (IBT #0210-001), and pooled serum from mice vaccinated with AP205wt (n = 3 mice/group) or AP205-LND (n = 3 mice/group). Assay performed in triplicate. A450 values were normalized to a total lysis control (RPMI+0.1% Triton-X114) set to 0% neutralization and a no lysis control (RPMI) set to 100% neutralization. One-way ANOVA *p* < 0.0001. Sidak’s multiple comparison test; ***, *p* < 0.001; ****, *p* < 0.0001. (**B**) Linear regression plot comparing lesion AUC for a 3-day Hla challenge of male BALB/c mice vaccinated with AP205-LND (n = 7 mice/group) or Qβ-LND (n = 6 mice/group) versus the log α-Hla titer, *p* = 0.0070. (**C**) ELISA showing Hla binding by antibodies from day 3 post-intoxication pooled lesion homogenate collected from male BALB/c mice previously vaccinated with AP205 (n = 8 mice/group), AP205-LND (n = 7 mice/group), Qβ (n = 7 mice/group), or Qβ-LND (n = 5 mice/group).

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
