# Peer review of "Vaccination with VLPs Presenting a Linear Neutralizing Domain of S. aureus Hla Elicits Protective Immunity"

_toxins, 2020, doi:10.3390/toxins12070450_

Round 1

Reviewer 1 Report

In the manuscript, authors describe the first VLP-based vaccines that target Hla, is a critical Staphylococcus aureus virulence factor. This is of relevance since no S. aureus or Hla vaccine has been successful in clinical trials. Specifically, they developed active VLP-based vaccines by displaying a 21 amino-acid Hla linear neutralizing domain (LND), is involved in protein oligomerization. They assessed vaccine efficacy using a murine skin challenge model. They also get into the mechanism of in vivo protection. The manuscript is well written and the experiments well designed. I have a minor comment and suggestion to improve the manuscript. Why did they select this murine skin model to assess the efficacy of the vaccine? I suggest to include this explanation in the text.

Author Response

Reviewer 1: "In the manuscript, authors describe the first VLP-based vaccines that target Hla, is a critical Staphylococcus aureus virulence factor. This is of relevance since no S. aureus or Hla vaccine has been successful in clinical trials. Specifically, they developed active VLP-based vaccines by displaying a 21 amino-acid Hla linear neutralizing domain (LND), is involved in protein oligomerization. They assessed vaccine efficacy using a murine skin challenge model. They also get into the mechanism of in vivo protection. The manuscript is well written and the experiments well designed. I have a minor comment and suggestion to improve the manuscript. Why did they select this murine skin model to assess the efficacy of the vaccine? I suggest to include this explanation in the text."

We appreciate the reviewer's comments. To address the issue of the selected mouse model, the following text has been added to page 4, lines 110-113 of the modified manuscript: “To assess the efficacy of our VLP-LND vaccines, we used a well-established mouse model of S. aureus skin infection/intoxication, where formation of a necrotic lesion is Hla-dependent [10, 19-21]. In this model, the necrotic lesion visibly forms on the skin surface and lesion size is easily measurable [22-26], allowing straightforward assessment of Hla vaccine efficacy.”

Reviewer 2 Report

The authors herei presented an interesting report on the use of viral like particle to induce immunity against a pathogen.

Even several effort have been done in this line, especially on the use of conserved S. aureus epitopes or virulence factor, to date no VLP-stretegies have been described for HLA.

An interesting approach was herein described and evaluated in terms of immunisation ability and neutralising activity.

The introduction sufficiently describe the state of art, results are clearly presented.

Also the use of E. coli strain, non-LPS free, in my opinion do not represent a relavant issue since the procedures applied for purifications. 

Author Response

Reviewer 2

"The authors herei presented an interesting report on the use of viral like particle to induce immunity against a pathogen.

Even several effort have been done in this line, especially on the use of conserved S. aureus epitopes or virulence factor, to date no VLP-stretegies have been described for HLA.

An interesting approach was herein described and evaluated in terms of immunisation ability and neutralising activity.

The introduction sufficiently describe the state of art, results are clearly presented.

Also the use of E. coli strain, non-LPS free, in my opinion do not represent a relavant issue since the procedures applied for purifications."

We greatly appreciate this review of our manuscript.